# Development of the Mesoscale Model GRAMM-SCI: Evaluation of Simulated Highly-Resolved Flow Fields in an Alpine and Pre-Alpine Region

Dietmar Oettl 

Air Quality Control, Government of Styria, Landhausgasse 7, 8010 Graz, Austria; dietmar.oettl@stmk.gv.at

**Abstract:** In this study, new developments implemented in the mesoscale model GRAMM-SCI are presented. GRAMM-SCI has been specifically developed for providing flow fields in the sub-kilometer range. A comprehensive model evaluation using wind, temperature, radiation, as well as soil moisture and soil temperature observations in an alpine valley and in a hilly pre-alpine region in Styria (Austria) is presented. Three one-way nested model domains were used, whereby the coarse model run (5000 m horizontal resolution) was initialized and forced using ERA5 reanalysis data. The grid sizes for the two inner domains were set to 1000 m and 200 m, respectively. Comparisons were carried out for a five-day period in October 2017 which was dominated by clear-sky conditions. Though, the observations reveal quite complex flow structures governed by interactions between synoptic flow and thermally-driven local flows, GRAMM-SCI was able to reproduce the main features satisfactorily. In addition, the new version of GRAMM-SCI shows significant improvement with regard to simulated air temperature compared with the previous one. Finally, microscale flow-field simulations were carried out for some monitoring sites that are apparently influenced by nearby buildings or vegetation.

**Keywords:** GRAMM-SCI; GRAL-SCI; flow-field modelling; mountain-valley winds; alpine meteorology; ERA5



## 1. Introduction

The main motivation for this study originates in the need for reliable wind fields for routine air-quality assessments in complex terrain. For instance, the Air Quality Control unit, Government of Styria, issues about 200 to 300 assessments per year for which pre-computed flow fields need to be readily available. This led to the development of so-called wind-field libraries established for a single calendar year (e.g., [1]), whereby the horizontal resolution is 200 m in the current library for the reference year 2015. Examples for wind-field libraries for varying purposes are reported for instance in [2–5], or [6]. The development of the mesoscale model GRAMM-SCI (Graz Mesoscale Model-Scientific) aims at providing such wind fields in highly complex terrain with a horizontal resolution below 1000 m. GRAMM-SCI ([7,8]) is a new branch of the model GRAMM, which has been further developed to make use of ERA5 (ECMWF Reanalysis 5th Generation) reanalysis data issued by the European Centre for Medium-Range Weather Forecasts (ECMWF, [9]). ERA5 is a global climate reanalysis dataset, currently covering the period from 1950 to the present on a spatial grid of 0.25 deg. Data is generally provided either at 37 pressure levels or at 137 model levels, the latter are used in GRAMM-SCI currently.

A previous publication by [8] provided a description of the new modelling features implemented in GRAMM-SCI and evaluated the model for the province of Styria (Austria) using results from the Weather Research and Forecasting model (WRF; [10]) as a benchmark and observations from numerous monitoring stations operated by the central weather service in Austria, ZAMG. One drawback of the study was that these monitoring stations are also used for establishing ERA5 reanalysis fields. In this work meteorological data

collected by the WegenerNet operated by the University of Graz [11] are mainly used for model evaluation. The main purpose of the WegenerNet is to provide high-density meteorological data for evaluating regional climate models [12]. The WegenerNet comprises two different regions: (1) Feldbach with a total of 155 monitoring stations within an area of about 22 km × 16 km, and (2) Johnsbach, where 11 meteorological sites are located within an area of 16 km × 17 km. Only a subset of all monitoring stations is equipped with anemometers. A detailed description of both areas is given in Section 3. Although, the focus of the study was placed on simulated surface winds, in addition the following parameters were evaluated when available: global and net radiation, air and soil temperature, and soil moisture.

## 2. Model Description

The Graz Mesoscale Model-Scientific (GRAMM-SCI) is a new branch of the public and open source version GRAMM 20.01 [13]. Since 2019 the GRAMM model has been further developed by the Air Quality Control unit of Styria, Austria, in order to make use of ERA5 reanalysis data for initialization and for prescribing transient boundary conditions. A comprehensive model description as well as first results of GRAMM-SCI for a region in the Eastern Alps were published in [8]. The following section highlights the changes in the model that have since been implemented.

While simulated surface winds in the study of [8] were in reasonable agreement with observations, the air temperature was significantly biased in the summer period. Thus, for improving modelled temperatures, the nudging technique previously solely applied for the horizontal flow-field components has also been implemented for the temperature, cloud-water content and humidity. For convenience, the methodology is briefly presented. Typically, a constant nudging term is applied in mesoscale models for the entire modelling domain, while the basic concept of the nudging technique in GRAMM-SCI follows the idea that the nudged model should have a maximum degree of freedom within the boundary layer, while above the boundary layer and close to the lateral boundaries large-scale meteorological fields should exhibit a stronger influence on the nudged model:

$$\psi_n = \psi_n - \alpha \left( \psi_n - \psi_{large-scale} \right) \tag{1}$$

$$\alpha = e^{-\gamma(z)} \tag{2}$$

In Equation (1) $n$ marks the actual time step and $\psi$ stands for one of the following quantities: horizontal wind-speed component, potential temperature, specific humidity, or cloud-water content. The nudging term $\alpha$ is height dependent as given in Equation (2). The exponent $\gamma(z)$ is 10 in the first model layer and 3 above the height $h_{crit}$, which is defined as the larger value of either 2000 m (or 3000 m during daytime conditions) above the surface or the maximum elevation within the modelling domain plus 500 m. The values for $\gamma(z)$ between the surface layer and $h_{crit}$ are linearly interpolated. In addition, $\gamma(z)$ is linearly decreased towards the lateral boundaries such that $\gamma(z)_{boundary} = 0.7\gamma(z)$.

Changes were also made regarding the calculation of the soil temperatures. ERA5 provides soil temperatures at several depths. GRAMM-SCI makes use of the top-soil as well as the "deep-soil" (~2 m depth) temperatures in the initialization procedure. The deep-soil temperature is kept constant throughout the simulation, while the soil temperatures above are calculated by means of eight soil layers [13]. Soil wetness is taken from ERA5 data and is kept constant—similar to the deep-soil temperature—between each re-initialization procedure.

Table 1 lists the most common CORINE land-use categories within the modelling domain and corresponding soil and surface characteristics used by GRAMM-SCI for the calculation of the surface-energy balance.

It is a common feature of mesoscale models to define thermodynamic variables as the sums of base-state variables and deviations from the base state (e.g., [14]). In the previous version of GRAMM-SCI problems were encountered in long model runs (simulations

periods of several weeks) that were related to the steady-state definition of the base temperature and humidity. Therefore, in the new version the base-state temperature and humidity are nudged towards ERA5 fields using the same nudging approach as described before. It is important to note that physical processes like gravity waves or deep convection are damped or even entirely prevented at higher altitudes where nudging towards large-scale fields becomes stronger.

**Table 1.** Land-use characteristics used in the Graz Mesoscale Model-Scientific (GRAMM-SCI) based on CORINE land-use categories.

| Code | Description | Albedo [-] | Emissivity [-] | Roughness Length [m] | Heat Conductivity [W/m/K] | Thermal Diffusivity [$m^2$/s] |
|---|---|---|---|---|---|---|
| 111 | Continuous urban fabric | 0.25 | 0.95 | 1 | 1 | $2.0 \times 10^{-6}$ |
| 112 | Discontinuous urban fabric | 0.25 | 0.95 | 0.5 | 1 | $1.3 \times 10^{-6}$ |
| 211 | Non-irrigated arable land | 0.19 | 0.92 | 0.1 | 0.2 | $7.0 \times 10^{-7}$ |
| 231 | Pastures | 0.19 | 0.92 | 0.1 | 0.2 | $7.0 \times 10^{-7}$ |
| 241 | Annual crops associated with permanent crops | 0.19 | 0.92 | 0.1 | 0.2 | $7.0 \times 10^{-7}$ |
| 243 | Land principally occupied by agriculture, with significant areas of natural vegetation | 0.19 | 0.92 | 0.2 | 0.2 | $7.0 \times 10^{-7}$ |
| 311 | Broad-leaved forest | 0.16 | 0.90 | 1 | 0.2 | $8.0 \times 10^{-7}$ |
| 312 | Coniferous forest | 0.12 | 0.90 | 1 | 0.2 | $8.0 \times 10^{-7}$ |
| 313 | Mixed forest | 0.14 | 0.90 | 1 | 0.2 | $8.0 \times 10^{-7}$ |
| 321 | Natural grasslands | 0.15 | 0.92 | 0.02 | 0.2 | $1.0 \times 10^{-6}$ |
| 332 | Bare rocks | 0.15 | 0.92 | 0.1 | 1 | $1.0 \times 10^{-6}$ |
| 511 | Water courses | 0.08 | 0.98 | 0.0001 | 100 | $1.0 \times 10^{-6}$ |

For the turbulence closure a zero-order mixing length model is utilized (e.g., [15]), where the turbulent viscosity depends on the local Gradient-Richardson number. The radiation model used in GRAMM-SCI is based on the work of [16]. Surface fluxes are calculated depending on stability and roughness-length according to relations proposed by [17]. Microphysical schemes for computing clouds and precipitation are not yet available in GRAMM-SCI. Cloud and snow cover are retrieved from ERA5 reanalysis data and are used diagnostically for adjusting the soil heat flux, albedo, and radiation [8].

While GRAMM-SCI has been driven by ERA5 reanalysis data using horizontal grid sizes between 3000 m to 250 m without using nesting methodologies by [8], in this work newly developed nesting and downscaling techniques for GRAMM-SCI were implemented and applied. In the largest model domain GRAMM-SCI is forced by ERA5 reanalysis data using a horizontal grid resolution of 5000 m (Figure 1). In the second, one-way nested domain, the horizontal resolution was 1000 m, and in the smallest domain a horizontal resolution of 200 m and sort of a dynamic-downscaling procedure [18] was applied. The nesting algorithm uses the same nudging technique as outlined before for the ERA5 reanalysis data, while in downscaling mode wind fields are hourly initialized and the modelling time is limited to 1200 s. The fundamental idea in the development of the downscaling

technique is that in the innermost modelling domain only atmospheric phenomena with length scales below the resolution of the nested model run (in this work: 1000 m) have to be considered. Typically, these are slope winds with length scales of a few hundred meters and time scales below one hour that would justify the limited modelling time of 1200 s in downscaling mode. It should be noted that also in downscaling mode the full set of prognostic equations is solved and the nudging technique is still applied. Indeed, the idea seems to be a novel approach in mesoscale modelling, and to the best of the author's knowledge no publication exists describing anything similar. The limited modelling time in downscaling mode naturally speeds up the simulation by a factor of three compared to a nested model run with the same horizontal resolution. For instance, the modelling times for the Feldbach region were 3 h, 8 h, and 10 h for the modelling domains utilizing 5000 m, 1000 m, and 200 m. In nesting mode 30 h simulation time would have been necessary for the 200 m resolution run.

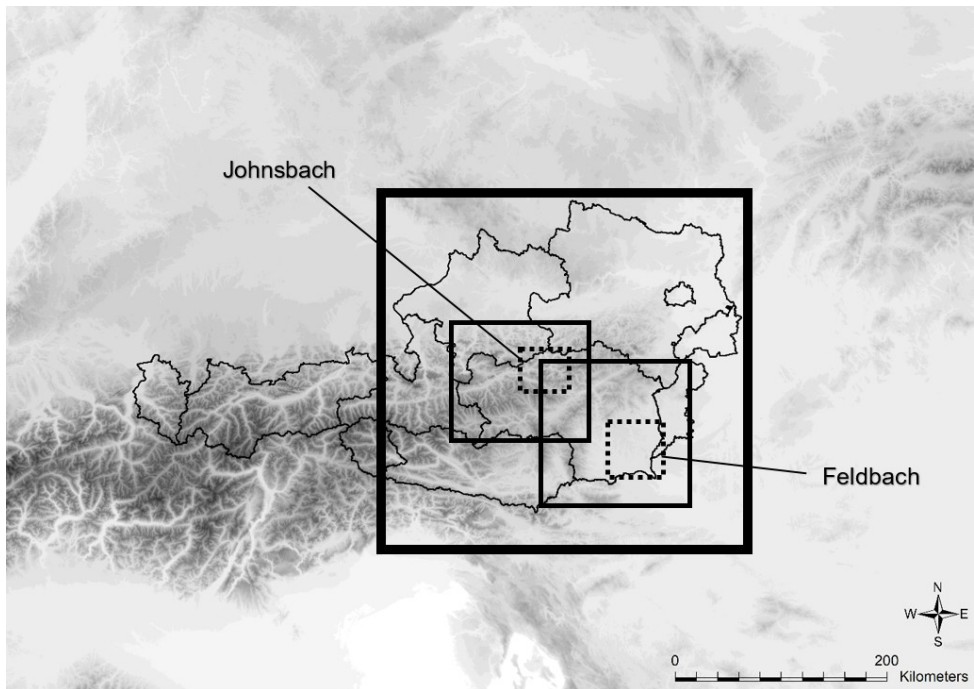

**Figure 1.** Locations of the modelling domains: domain 1 (5000 m horizontal resolution, thick black line), two nested domains (1000 m horizontal resolution, thin black lines), and two downscaling domains (200 m horizontal resolution, dotted lines).

ERA5 data based on 6 h intervals were used in this work and boundary conditions were updated accordingly. An additional simulation was carried out in which ERA5 data was interpolated on the GRAMM-SCI grid (500 m horizontal resolution), but without solving the conservation equations in order to demonstrate the quality of the ERA5 reanalysis data and to evaluate the added value of applying GRAMM-SCI. In all simulations 26 layers were defined in the vertical direction with 18 of them in the lowest 3 km and the first grid point 5 m above the surface. The model top was set at 18 km.

For several selected monitoring stations, which seemed to be influenced by local buildings or vegetation, microscale flow-field simulations were carried out by means of the prognostic model implemented in GRAL-SCI (Graz Lagrangian Model-Scientific; [19]) using a horizontal resolution of 3 m. The coupling methodology between GRAL-SCI and GRAMM-SCI has already been published in [20] and is not outlined in this work. It should be noted that GRAL-SCI is a hybrid model combining the aforementioned non-hydrostatic microscale flow-field model with a Lagrangian dispersion model. The model is available as open source code under the GPL3.0 license on request.

### 3. Study Area

Figure 1 illustrates the location of the modelling domains. The largest domain (5000 m horizontal resolution) extends to about 300 km × 300 km and covers the eastern Alps in Austria. The highest mountains exceed 3000 m. To the southeast an Alpine foreland stretches out, which is characterized by hilly terrain with heights up to 600 m above sea level. The two nested domains (1000 m horizontal resolution) cover areas of 130 km × 100 km (Johnsbach) and 180 km × 140 km (Feldbach), respectively. Downscaling (200 m horizontal resolution) was applied for the two smallest domains, which are indicated by dotted lines in Figure 1. The Feldbach domain has a size of 41 km × 43 km and the Johnsbach domain of 28 km × 22 km.

Typically, diurnal mountain winds develop during fair weather conditions (e.g., [21]). Indeed, valley-wind systems, mountain-plain wind systems, or slope winds (e.g., [22]) can be observed frequently in the study area in such conditions. In order to investigate the capability of GRAMM-SCI to reproduce the local wind systems in the area, the period from 13 to 18 October 2017 was selected. As can be seen from the weather map, issued by the Austrian Meteorological Service ZAMG for the 13 October 2017 (Figure 2), weak pressure gradients prevailed at sea level during this period.

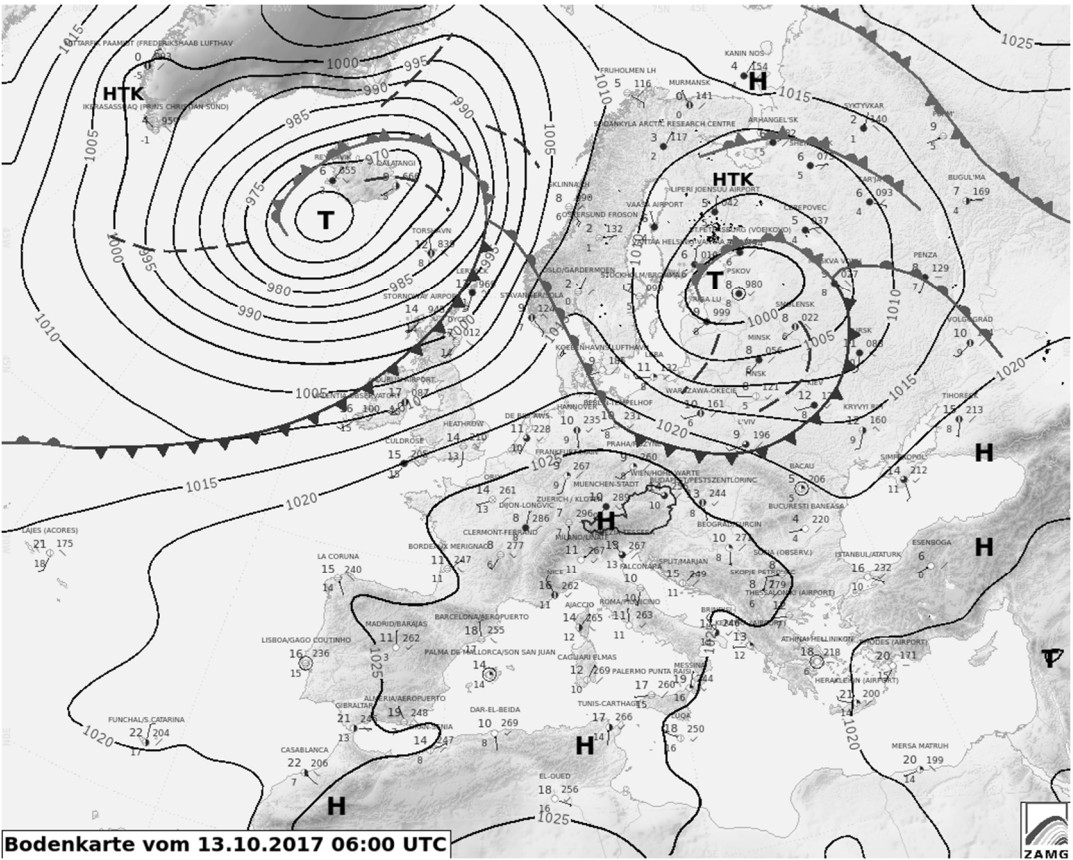

**Figure 2.** Weather map issued by the Austrian Meteorological Service ZAMG for the 13 October 2017 for 6 UTC [23]. H = high pressure; T = low pressure; HTK = upper-level low.

The location of the monitoring sites used for model evaluation are depicted in Figure 3 for the Johnsbach and for the Feldbach domain. The Johnsbach area covers an Alpine valley with altitudes between 700 m and 2000 m, whereas the Feldbach region is characterized by gently rolling hills and valleys with a typical depth of 100 m to 300 m. The observed parameters at each station used for model evaluation are listed in Table 2. Modelled wind speeds and directions were evaluated at the height of each wind sensor by applying a linear

interpolation between neighboring grid cells. Soil temperature and moisture measurements are registered at 0.2 m depth.

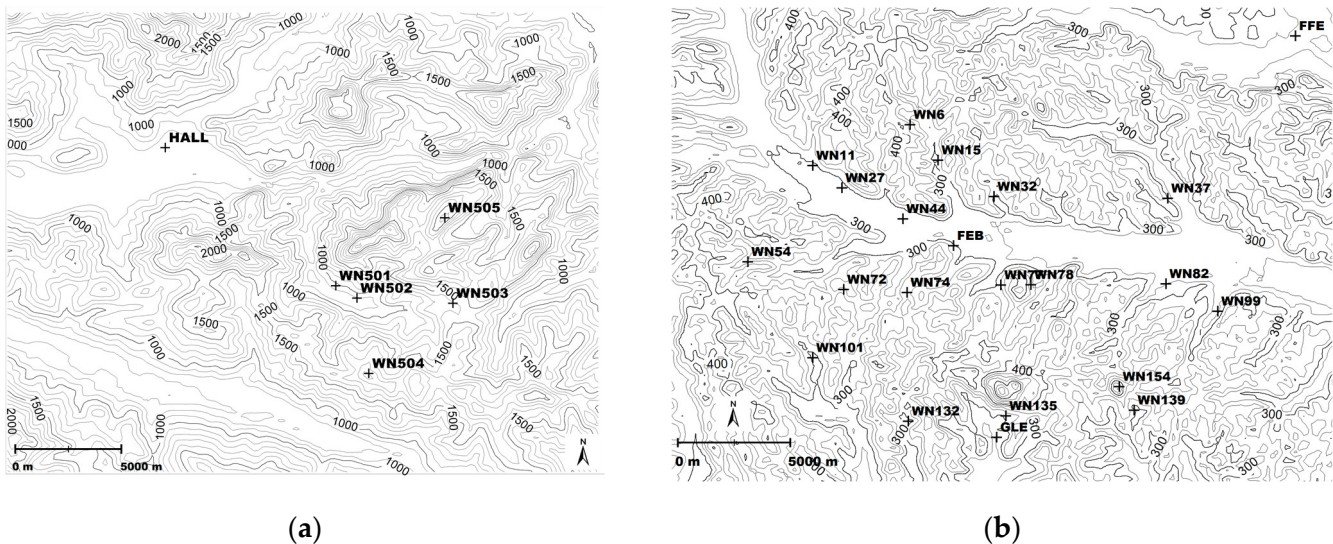

(**a**)  (**b**)

**Figure 3.** Location of the monitoring sites in (**a**) the Johnsbach (100 m isopleths) and (**b**) Feldbach (25 m isopleths) modelling domains.

**Table 2.** Monitoring sites used for model evaluation and observed parameters (u = wind, $T_{air}$ = air temperature, $T_{soil}$ = soil temperature, $M_{soil}$ = soil moisture, $r_{net}$ = net radiation, $r_{glob}$ = global radiation). "WNxxx" stations belong to the WegenerNet, the other stations are operated by the central weather service ZAMG.

| Station | Longitude (°) | Latitude (°) | Altitude (m) | Parameters | Wind Sensor Height above Ground (m) |
|---|---|---|---|---|---|
| GLEICHENBERG (GLE) | 15.90361 | 46.87222 | 269 | $u, T_{air}$ | 18 |
| FELDBACH (FEB) | 15.87972 | 46.94889 | 323 | $u, T_{air}$ | 10 |
| FUERSTENFELD (FFE) | 16.08083 | 47.03083 | 271 | $u, T_{air}, r_{glob}$ | 10 |
| WN6 | 15.85507 | 46.99726 | 398 | $T_{soil}, M_{soil}$ | - |
| WN11 | 15.79801 | 46.98137 | 300 | $u, T_{air}$ | 10 |
| WN15 | 15.87115 | 46.98299 | 297 | $T_{soil}, M_{soil}$ | - |
| WN27 | 15.81499 | 46.97232 | 298 | $T_{soil}, M_{soil}$ | - |
| WN44 | 15.85036 | 46.95979 | 288 | $u, T_{air}$ | 55 |
| WN54 | 15.75960 | 46.94327 | 348 | $T_{soil}, M_{soil}$ | - |
| WN72 | 15.81543 | 46.93182 | 337 | $u, T_{air}$ | 18 |
| WN77 | 15.90706 | 46.93294 | 306 | $T_{soil}, M_{soil}$ | - |
| WN78 | 15.92462 | 46.93291 | 372 | $T_{soil}, M_{soil}$ | - |
| WN82 | 16.00336 | 46.93263 | 276 | $u, T_{air}$ | 10 |
| WN99 | 16.03337 | 46.92135 | 270 | $T_{soil}, M_{soil}$ | - |
| WN101 | 15.79682 | 46.90473 | 304 | $u, T_{air}$ | 14 |
| WN132 | 15.85215 | 46.87902 | 295 | $u, T_{air}$ | 10 |
| WN139 | 15.98394 | 46.88229 | 307 | $u, T_{air}$ | 10 |
| WN154 | 15.97547 | 46.89181 | 471 | $u, T_{air}$ | 10 |
| HALL | 14.49083 | 47.59472 | 637 | $u, T_{air}$ | 10 |
| WN501 | 14.59800 | 47.53640 | 920 | $u, T_{air}, r_{net}, r_{glob}$ | 10 |
| WN502 | 14.61130 | 47.53120 | 860 | $u, T_{air}, r_{net}, r_{glob}$ | 10 |
| WN503 | 14.67132 | 47.52922 | 1344 | $u, T_{air}, r_{net}, r_{glob}$ | 10 |
| WN504 | 14.61885 | 47.49936 | 1969 | $u, T_{air}, r_{net}, r_{glob}$ | 6 |
| WN505 | 14.66605 | 47.56540 | 2191 | $u, T_{air}, r_{net}, r_{glob}$ | 6 |

## 4. Results

The following metrics were used for evaluating model results, whereby *O* stands for observation, *M* for measured values, and *n* for the number of data points.

Pearson correlation coefficient (*R*):

$$R = \frac{\sum_{i=1}^{n}(M_i - \overline{M})(O_i - \overline{O})}{\sqrt{\sum_{i=1}^{n}(M_i - \overline{M})^2}\sqrt{\sum_{i=1}^{n}(O_i - \overline{O})^2}} \tag{3}$$

Mean error:

$$BIAS = \frac{1}{n}\sum_{i=1}^{n}(M_i - O_i) \tag{4}$$

Root mean square error:

$$RMSE = \sqrt{\frac{1}{n}\sum_{i=1}^{n}(O_i - M_i)^2} \tag{5}$$

Mean wind direction error:

$$BIAS_{dir} = \frac{1}{n}\sum_{i=1}^{n}\{arccos[cos(M_i - O_i)]\} \tag{6}$$

In some studies, the overall mean of observed quantities is used for demonstrating mesoscale model performance. This seems to be particularly useful as long as the horizontal resolution in the modelling is still rather coarse ($\geq$1 km). However, the fine horizontal resolution used in this study allows for direct comparison with observations at specific locations, which was therefore the preferred method. Unless otherwise stated only results for the innermost model domains with a horizontal resolution of 200 m are presented.

### 4.1. Radiation

Examples for modelled and observed global and net radiation are presented in Figure 4. On the 13 October GRAMM-SCI clearly underestimates the global radiation in the Feldbach domain. As mentioned in Section 2, cloud formation is not yet calculated in GRAMM-SCI. Therefore, the underestimation of global radiation can only be due to an overestimated cloud cover by ERA5. However, GRAMM-SCI has a general tendency for underestimating the global radiation at the site FFE in the Feldbach domain even on the following days where clear-sky conditions prevailed. On the other hand, good agreement is found at site WN501 in the Johnsbach valley for the whole simulation period. A large bias is found for the station WN505 ($-71$ W/m$^2$). The station is located on a mountain peak at 2191 m making the simulated global radiation very sensitive to the exact position of the station within the model's topography. That means that even a slight dislocation between the station and the mountain peak as mapped by the model immediately leads to an erroneous orientation of the model's surface. Quite good results were obtained for the net radiation at the monitoring site WN505, while within the Johnsbach valley GRAMM-SCI apparently tends to overestimate the outgoing longwave radiation as the radiation in nighttime is negatively biased at the site WN501. The *BIAS* over all stations is $-16$ W m$^{-2}$ for the net radiation and $-39$ W m$^{-2}$ for the global radiation indicating that the radiation model as well as the empirical procedure of taking into account cloudiness as outlined in [8] works properly in the prevailing weather conditions during the investigated period.

### 4.2. Soil Temperature and Moisture

Figure 5 depicts the observed and simulated soil temperatures and soil-moisture contents for two monitoring sites in the Feldbach domain. Note that no data was available for the Johnsbach region. Modelled soil temperatures were calculated by using a linear interpolation between the fifth and sixth soil layer located 0.18 m and 0.42 m below the

surface. While GRAMM-SCI suggests weak spatial variation of average soil temperatures at 0.2 m depth (less than 0.5 K difference among all sites), observed temperatures in contrast vary by about 2 K over all stations. This is an indication that the spatial variability is largely influenced by local variations in the soil texture, which cannot be reflected by assigning default soil properties based on CORINE land-use categories (Table 1). Nevertheless, the model is able to reproduce the damped daily oscillation of the soil temperature. On average GRAMM-SCI tends to overestimate the soil temperatures by 1.3 K. The soil-moisture content is not explicitly calculated by the model, but is only interpolated from ERA5 reanalysis data. However, the values agree rather well with the observed ones. A slight negative *BIAS* of −0.07 between modelled and observed values was found. It is evident from the observations that there is little spatial variation of the soil-moisture content in the Feldbach domain, which is probably caused by the fact that no precipitation occurred during the investigated period. Therefore, the soil-moisture content provided by ERA5 on a grid size of about 19 km at this latitude is sufficient in such conditions.

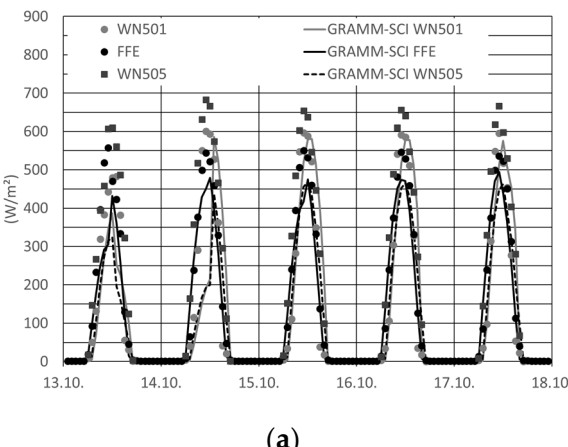

(**a**)

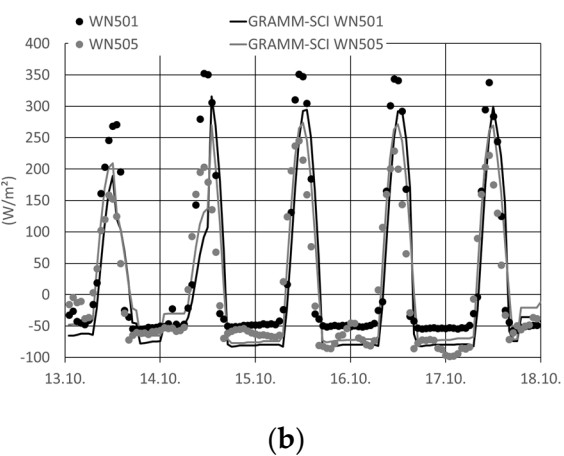

(**b**)

**Figure 4.** Observed and modelled global radiation (**a**) and net radiation (**b**) at selected sites.

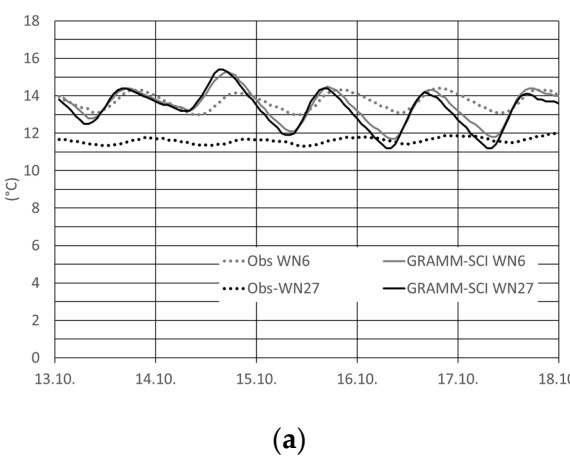

(**a**)

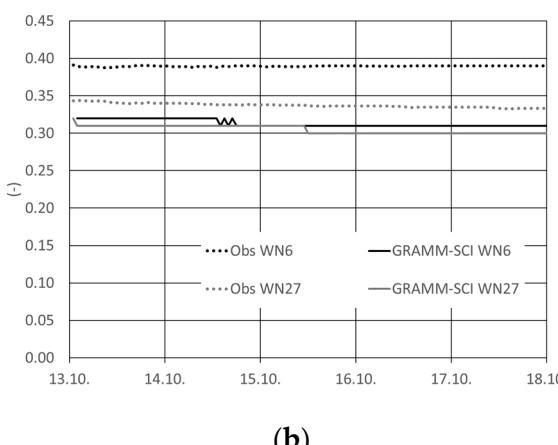

(**b**)

**Figure 5.** Observed and modelled soil temperatures (**a**) and soil-moisture contents (**b**) at selected sites in the Feldbach area.

### 4.3. Air Temperature

As already outlined in Section 2, in the present version of GRAMM-SCI, temperatures are nudged towards ERA5 reanalysis data. The *BIAS* for the entire period and over all stations is 0.6 K, which is in the range found for other existing modelling studies performed in Alpine regions (e.g., [24–27]). Furthermore, it is a significant improvement compared to the study of [8], who reported a mean error of about −3.0 K for an application of

GRAMM-SCI in an eastern Alpine region for a summer week. It should be noted that these simulations have been repeated with the present model version of GRAMM-SCI resulting in a *BIAS* of only −0.1 K. Hence, the introduction of nudging the air temperature as well as the base-state temperature for avoiding long-term drifts of the model significantly improves model results in this respect. Yet, at some locations large biases are found (Figure 6). For instance, at the site Hall in the Johnsbach domain (Figure 3) the temperature bias is 3 K, which is probably caused by the overestimation of the wind speeds related to a nocturnal drainage flow preventing the development of stagnant conditions and related cold-air pools (Figure 9 bottom right).

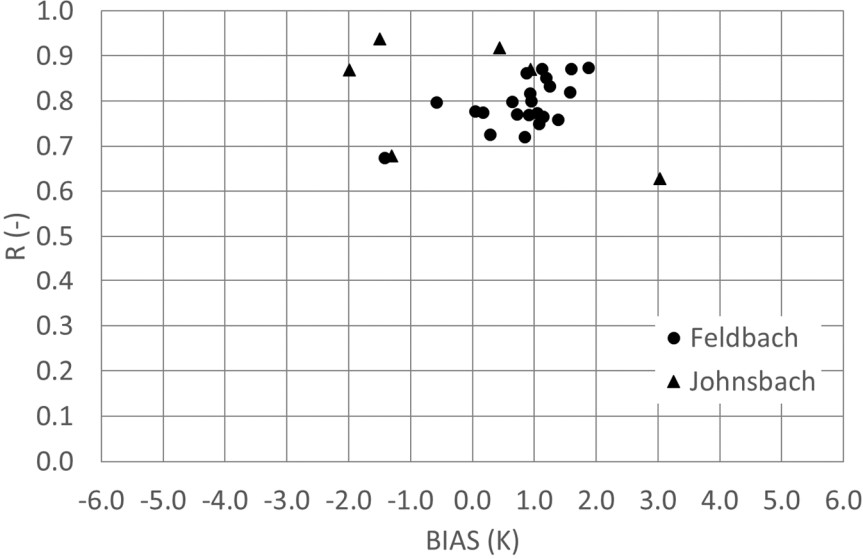

**Figure 6.** *BIAS* and correlation coefficient *R* of modelled air temperatures in the Feldbach and Johnsbach valley domains.

　　　　Examples for simulated and observed temperatures in the two domains are shown in Figure 7. It is worthwhile noting that the daily temperature variation and the absolute maxima/minima are fundamentally the same in both regions, even though the Johnsbach valley is located at higher altitudes than the stations in the Feldbach area. GRAMM-SCI reproduces these features quite well. It is remarkable that the significant nocturnal temperature differences between the stations WN501 and WN502 in the Johnsbach valley are captured by the model although the horizontal distance between the two stations is only 1.1 km and both stations are located within the same valley (Figure 3). The significant overestimation of maximum daytime temperatures between 15 and 18 October at the site WN501 points to some local effects in the simulations requiring further investigations. In the Feldbach domain cooling during the night between 13 and 14 October is not well modelled as ERA5 reanalysis data suggest an overcast sky, while clear-sky conditions prevailed.

### 4.4. Wind Speed and Direction

　　　　A few studies based on the monitoring stations of the WegenerNet were carried out in the past. [28] developed the empirical diagnostic tool WPG (Wind Product Generator) for analyzing flow fields with a high spatial and temporal resolution. In subsequent works, the diagnostic INCA model [29], which is coupled with the numerical weather prediction model ALARO [30], and the numerical weather prediction model COSMO-CLM [31] were evaluated against gridded wind fields calculated with the WPG tool [12]. Hence, the performance of INCA and COSMO-CLM reported in these studies is not comparable with the model quality indicators used in this work in a straight forward manner, because GRAMM-SCI outputs are compared with observed winds paired in space and time. Moreover, the evaluation periods in this study differ from those used in these

previous studies. Having said that, INCA and COSMO-CLM model results are partly included in the discussion herein as they are a kind of benchmark regarding wind-field modelling for the two modelling domains.

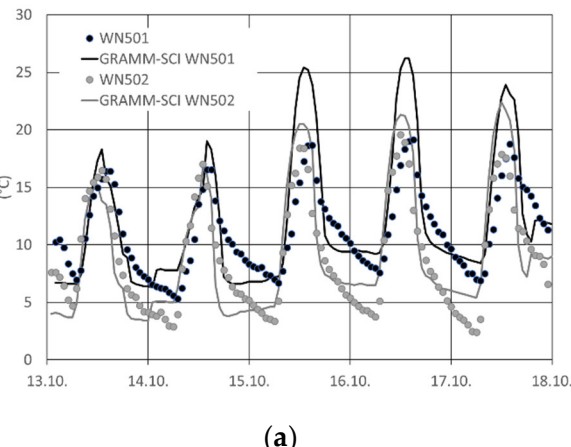
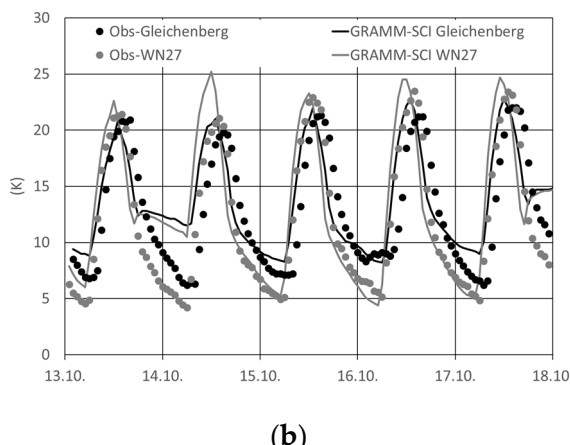

(**a**)                                                                                       (**b**)

**Figure 7.** Observed and modelled air temperatures at selected sites in the Johnsbach (**a**) and Feldbach (**b**) domains.

Figure 8 depicts the *BIAS* and *RMSE* for all monitoring sites in the two domains regarding wind speed and direction. The European Environmental Agency (EEA) suggests a mean $BIAS < \pm 0.5\,\mathrm{m\,s^{-1}}$ and a $RMSE < 2\,\mathrm{m\,s^{-1}}$ as benchmarks for model evaluation [32], whereas the United States-Enivronmental Protection Agency (US-EPA) differentiates between "simple" and "complex" conditions when applying quality criteria [5]. For "simple" conditions the same thresholds are used as in [32], while for "complex" conditions $1.5\,\mathrm{m\,s^{-1}}$ for the *BIAS*, and $2.5\,\mathrm{m\,s^{-1}}$ for the *RMSE* are utilized. Both model quality objectives are indicated in Figure 8 (left panel). The wind speeds in the Feldbach region are much better captured by the model than in the Johnsbach domain, which is mainly due to the fact that wind speeds were quite high at the mountain peaks in the Johnsbach area. In contrary, model performance regarding wind direction is better in the Johnsbach domain.

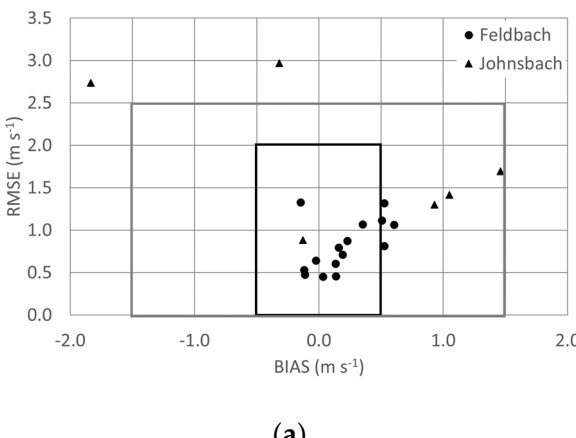
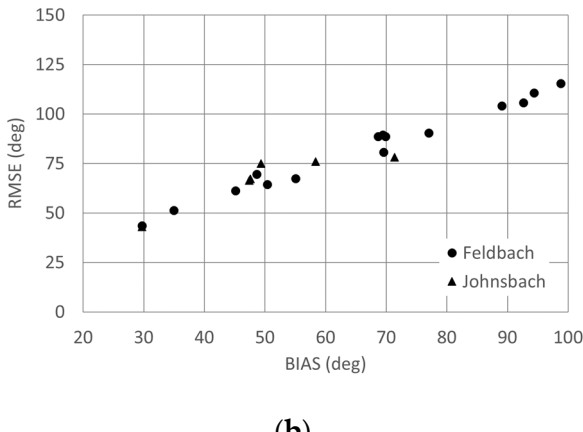

(**a**)                                                                                       (**b**)

**Figure 8.** *BIAS* and *RMSE* of modelled wind speeds and directions in the Feldbach and Johnsbach domains. The rectangles mark the quality criteria of the European Environmental Agency (EEA) (black) and United States-Environmental Protection Agency (grey) for "complex" conditions.

Some performance indicators are listed in Table 3 including the model run with 1000 m horizontal resolution and a simulation, where ERA5 reanalysis data has been interpolated on a GRAMM-SCI grid with 500 m resolution without solving the conservation equations.

The hit rates for wind speed refer to the quality objective of the [32], while the corresponding hit rates for wind direction are based on the suggested quality criteria of [8]

$$BIAS \leq \frac{46}{Max(\overline{u}, \ 0.5)} + 25 \qquad (7)$$

where $\overline{u}$ is the mean wind speed at a monitoring station over the entire period. As expected, the model runs with the highest horizontal resolution are performing best. However, it should be stressed that simulations using small horizontal grid sizes are often prone to numerical instabilities (e.g., [33]) and do not necessarily improve results in general (e.g., [18]). It appears that the methodologies implemented in GRAMM-SCI for keeping simulations numerically stable [8] function satisfactorily in the model runs presented herein.

**Table 3.** Statistical performance indicators for wind speed and direction for the different model runs. Corresponding best-performance indicators are printed bold.

| | | BIAS | | RMSE | | R | | Hit Rates | |
|---|---|---|---|---|---|---|---|---|---|
| | | u (m s$^{-1}$) | $\Theta$ (deg) | u (m s$^{-1}$) | $\Theta$ (deg) | u (-) | $\Theta$ (-) | u (-) | $\Theta$ (-) |
| ERA5 | Johnsbach | 1.21 | 74 | 2.04 | 87 | **0.40** | 0.36 | 0.0 | 0.33 |
| | Feldbach | 0.82 | 80 | 1.23 | 94 | **0.32** | 0.17 | 0.2 | 0.67 |
| 1000 m | Johnsbach | 2.01 | 60 | 2.77 | 76 | 0.27 | 0.22 | 0.0 | 0.50 |
| | Feldbach | **0.25** | 73 | **0.79** | 88 | 0.25 | 0.19 | **0.93** | 0.67 |
| 200 m | Johnsbach | **0.95** | **51** | **1.83** | **68** | 0.36 | **0.45** | **0.17** | 0.67 |
| | Feldbach | **0.25** | **66** | 0.82 | **82** | **0.32** | **0.20** | 0.73 | **0.80** |

A comprehensive literature study about typical performances of mesoscale models was undertaken by [34], who evaluated reported values of *R*, *BIAS*, and *RMSE* for a large number of model results. For example, the *RMSE* for the best 50% model results was found to be 1.8 m s$^{-1}$ and 72 deg for wind speed and direction, respectively. The model performance of GRAMM-SCI regarding wind speed is considerably better, while the quality regarding wind direction for the Feldbach domain is slightly worse. At this stage it should be stressed that predicting wind direction is particularly difficult in low-wind speed, stagnant meteorological conditions (e.g., [25,27,35,36]), which also becomes evident when comparing the model performance for the Johnsbach and Feldbach regions. The wind speed averaged over all stations and the entire period was 2.4 m s$^{-1}$ in the Johnsbach domain but only 0.76 m s$^{-1}$ in the Feldbach area, which is likely the explanation for the worse performance concerning wind direction in the Feldbach domain. In contrary, the *BIAS* and *RMSE* with regard to wind speed typically increase with higher wind speeds.

Schlager, C., et al. [12] compared the models INCA and COSMO-CLM with the output of the WPG tool, which provides gridded wind data based on the observations in the Feldbach and Johnsbach areas. For weather conditions that favor the development of thermally induced circulations a *BIAS* of 0.34 m s$^{-1}$/38 deg. and 1.32 m s$^{-1}$/55 deg. was obtained for the INCA and COSMO-CLM model for the Feldbach region, respectively. For the Johnsbach area only INCA was compared with WPG and resulted in a *BIAS* of $-1.37$ m s$^{-1}$/68 deg. Although care has to be taken when comparing these model quality indicators with the ones obtained for GRAMM-SCI, one can safely state that results are encouraging. The circumstance that the COSMO-CLM model, which was utilized with a horizontal grid resolution of 3 km performs better with regard to the wind-direction (*BIAS*) in the Feldbach region than GRAMM-SCI using a resolution of 200 m, is probably due to the underlying comparison with gridded data from WPG and not with the observations themselves. As mentioned before, observed wind directions often exhibit a large variability in low-wind speed conditions, which cannot be captured by mesoscale models (e.g., [37]).

Therefore, it might be the case that the WPG model is spatially smoothing observed low-frequent meandering motions leading to better agreements with the output from mesoscale models than if they were compared solely with observations.

Some examples of observed and modelled winds are illustrated in Figure 9. The left panel shows results for the Johnsbach domain and the right panel for the Feldbach region. Even though the whole period was determined by weak synoptic forcing at sea level, it is interesting to note that the monitoring station WN505 located on a mountain peak (2191 m) registered quite high wind speeds between 7 to 15 m s$^{-1}$ on 13 and 14 October. These strong north-westerly winds are well simulated by GRAMM-SCI, which is mainly due to the high quality of the ERA5 reanalysis data in this case. At the same time local thermally-driven winds developed as observed at the monitoring sites WN501 and HALL for example. The measured wind speeds were below 2 m s$^{-1}$ at the station WN501 (920 m) and even stagnant at the site HALL during the night. It should be mentioned that wind speed and direction are automatically labelled zero by the Austrian weather service ZAMG as soon as the wind speed falls below the instrument's detection limit. In such cases the wind direction is removed from the dataset. Figure 10 depicts the observed and modelled wind for 13 October at 7 p.m. The cold-air drainage flow in the Johnsbach valley with easterly wind directions is nicely visible as well as the northerly downslope winds at the station HALL, and the strong north-westerly synoptic-scale flow at the mountain crests. GRAMM-SCI is able to capture these complex flow regimes well.

The top-right panel in Figure 9 shows the observed and modelled wind at the station WN154 in the Feldbach domain, which is situated on top of a hill (471 m) approximately 150 m above the surroundings. One of the peculiarities at this station is the fact that—in agreement with most of the stations—southerly wind directions are registered during the day. This up-valley wind is caused by the main Alpine ridge approximately 50 km north-west of the monitoring station (Figure 1) and is hardly influenced by the local topography. On the contrary, registered nocturnal wind directions at this station do not follow any pattern, indicating that the site is not affected by a regular down-valley wind. Modelled winds match observations quite satisfactorily at this site suggesting that GRAMM-SCI is able to reproduce the interaction between the thermally-induced winds by the main Alpine ridge and its forelands in the southeast, and also the large-scale winds that are influenced by the main Alpine ridge. Note, that the large-scale wind in the Johnsbach domain was from the north-west on 13 and 14 October, while the corresponding wind directions at the site WN154 were from the south and south-east during this time.

A nice example of a thermally-driven local wind in a small side valley is visible at the monitoring site WN101 (Figure 9, mid plot in the right panel). The side valley extends approximately 3 km in the north-south direction whereby the valley floor is less than 500 m wide. The observed diurnal variation in wind direction and speed is a textbook example for a local valley-wind system which is almost perfectly captured by the model. Figure 11 illustrates the complexity of the observed and modelled nocturnal drainage flows. Practically each valley generates its own down-valley wind. However, as soon as wind speeds become very low, one observes low-frequent meandering motions. For instance, the registered wind direction at the station WN135 indicates an up-valley wind in Figure 11, which is likely due to flow meandering as the observed wind speed was only 0.07 m s$^{-1}$ at this time.

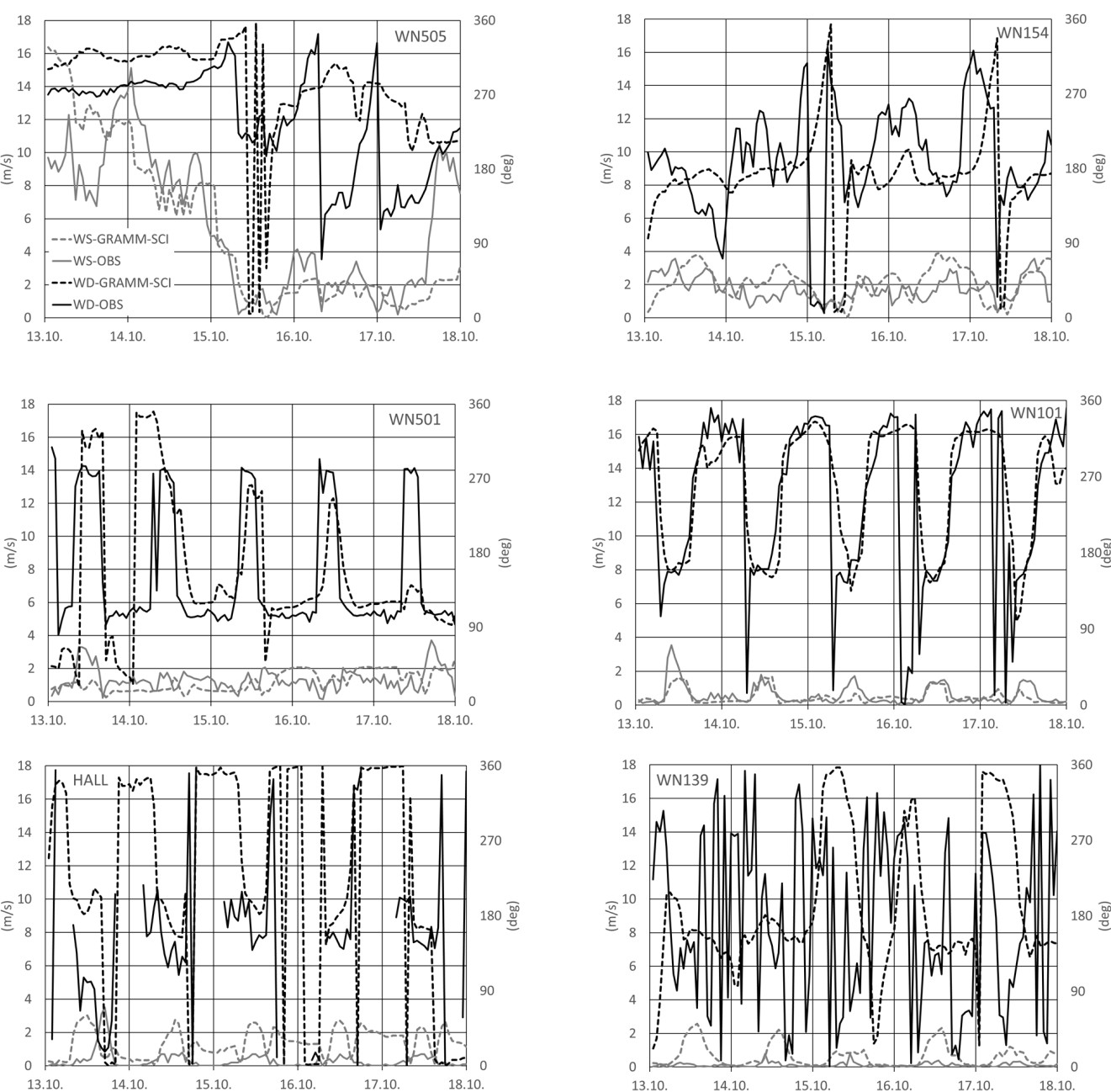

**Figure 9.** Observed and modelled wind speed and direction at selected sites in the Johnsbach (**left**) and Feldbach (**right**) domains. Dotted lines refer to GRAMM-SCI results, full lines to observations, black refers to wind direction, and grey to wind speed.

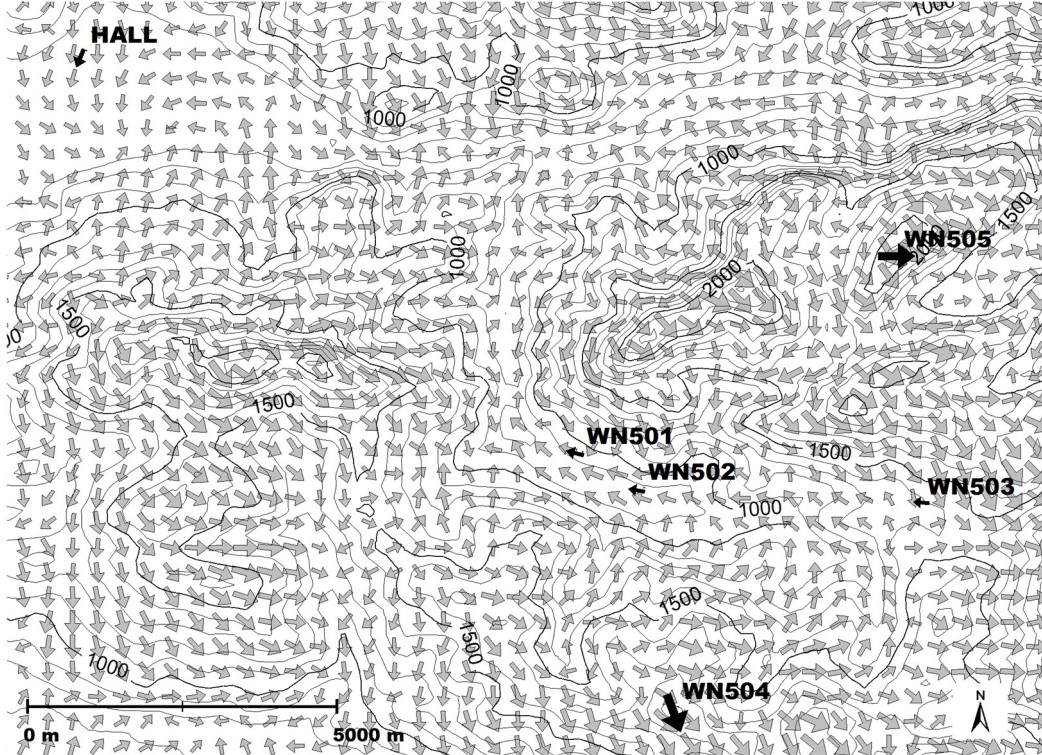

**Figure 10.** Simulated (grey) and observed (black) flow in the Johnsbach domain for 13 October 2017 at 7.p.m. (only every second modelled vector is shown).

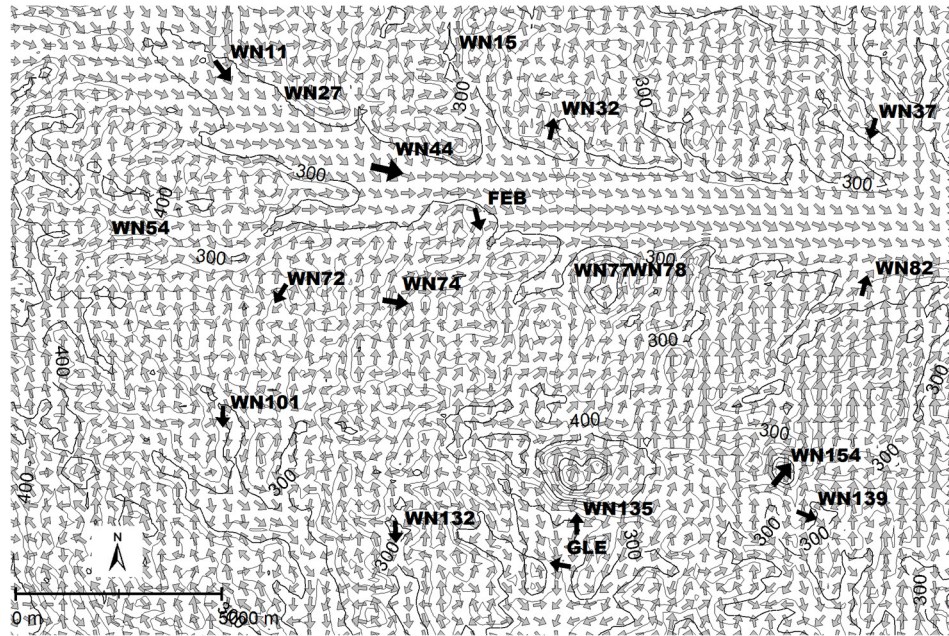

**Figure 11.** Simulated (grey) and observed (black) flow in the Feldbach domain for 16 October 2017 at 3.a.m. (only every second modelled vector is shown).

Similarly, measured wind speeds at the site WN139 (Figure 9, bottom-right panel) were extremely low during the entire period resulting in an average of solely 0.13 m s$^{-1}$. Corresponding wind directions were quite irregular indicating the presence of low-frequent meandering motions throughout the time. With our current understanding of these low-

oscillating mesoscale motions it is hard to conceive that any kind of currently available state-of-the art model would be able to capture this atmospheric phenomenon in detail. GRAMM-SCI is neither able to simulate hourly wind directions well nor does it match the observed wind speeds during the daytime. Especially, on 13, 14, and 16 October, when the prevailing wind direction was from the south, modelled wind speeds are clearly overestimated. A closer examination of the site revealed a possible influence of a few trees south of the station at a distance of about 17 m. While the height of the anemometer is 10 m above ground level, the broad-leaved trees are up to 15 m high according to the available airborne laser-scan data for Styria (https://www.landesentwicklung.steiermark.at/cms/ziel/141976122/DE/ last accessed 20 January 2021). A microscale simulation was carried out by means of the prognostic, non-hydrostatic flow-field model implemented in the Lagrangian particle model GRAL-SCI (e.g., [19]). The model is able to take vegetation into account [38] and has been evaluated using wind observations within an Aspen canopy layer [39]. A detailed description of the model and the nesting technique used for coupling with GRAMM-SCI would go beyond the scope of this work. The reader is referred to the comprehensive descriptions provided in [19] and [38] for more information. Basically, the computed wind field with GRAMM-SCI is interpolated on the Cartesian grid used by GRAL-SCI. By keeping the inflow boundary wind constant, the model is run until a steady-state flow field is obtained. This procedure is repeated for each hour of the investigated period.

Figure 12 depicts the resultant wind speeds with GRAL-SCI utilizing a grid size of 3 m $\times$ 3 m $\times$ 1 m and a leave-area density of 0.5 m$^2$ m$^{-3}$, which is the default value for this type of vegetation in the model. The model domain was 300 m $\times$ 300 m $\times$ 50 m. As expected, simulated wind speeds fit much better to observations confirming that the vegetation forms a barrier for southerly winds. The *BIAS* reduces from 0.52 m s$^{-1}$ to 0.32 m s$^{-1}$ by applying the microscale model. An example of a computed microscale flow field at a height of 10 m above ground level is illustrated in Figure 13. The influence of the trees on the flow field is clearly visible, while the buildings do not alter the flow as they are lower (6 m roof top) than the height of the depicted wind field. Moreover, it should be noted that the wind direction in the wake of the vegetation does not change significantly. Thus, GRAL-SCI suggests a significant drop in wind speed but no recirculation zone with opposing wind directions, which agrees with the observation. Further simulations were carried out for the monitoring stations WN72 and WN82, which are also influenced by nearby buildings. For these stations the *BIAS* reduced from 0.53 m s$^{-1}$ to $-0.17$ m s$^{-1}$, and from 0.60 m s$^{-1}$ to 0.39 m s$^{-1}$, respectively.

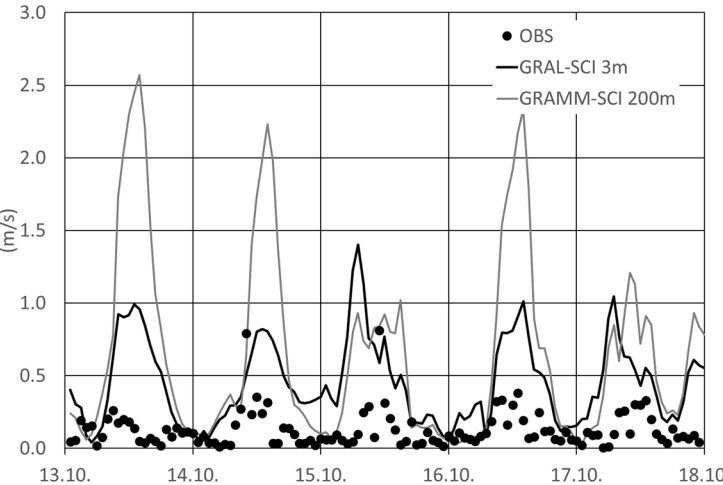

**Figure 12.** Observed and modelled wind speeds at the monitoring station WN139 in the Feldbach domain using the microscale flow-field model implemented in GRAL-SCI.

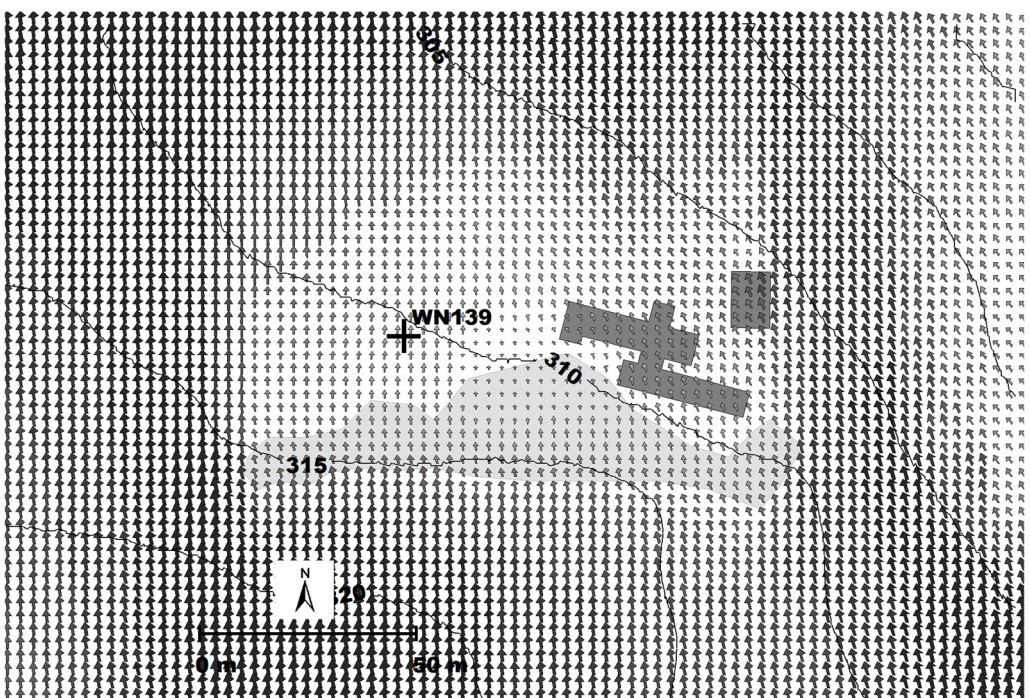

**Figure 13.** Example for a calculated microscale flow field 10 m above ground level for the 13 October 2017, 2.p.m., utilizing GRAL-SCI. The vegetated area is marked with light grey and buildings by grey colors. Topography is indicated as black lines.

## 5. Conclusions

The study demonstrates the ability of the mesoscale model GRAMM-SCI to simulate mountain-wind systems and their interaction with synoptic-scale flows in highly complex terrain. Especially the newly implemented nudging technique for potential temperature and the base-state temperature, improved modelled air temperatures significantly and may prevent long-term drifts in the simulations. Furthermore, the applied nesting and the novel downscaling methodologies, being developed in the scope of this work, were proven to provide numerically robust results within reasonable computation times. In particular, the downscaling approach used for the innermost domains utilizing very small horizontal grid sizes is three times faster compared to nesting. Distributed computation on different computers is readily possible in downscaling mode, because the model is re-initialized each hour and hence, each model run is completely independent of others.

It was shown that in combination with GRAL-SCI even buildings and vegetation can be taken into account, such that flow-fields over a large range of scales can be provided in highly complex terrain. Bearing in mind that high-resolution mesoscale flow-field simulations in Alpine areas are still few in number, the performance of GRAMM-SCI coupled with ERA5 reanalysis data is encouraging. Nevertheless, the focus of the evaluation of GRAMM-SCI so far was on fair-weather conditions with rather weak synoptic forcing. Therefore, further research will be necessary for situations with strong large-scale forcing, overcast skies, or existing snow cover.

**Funding:** This research received no external funding.

**Institutional Review Board Statement:** Not applicable.

**Informed Consent Statement:** Not applicable.

**Data Availability Statement:** Publicly available datasets were analyzed in this study. This data can be found here: https://wegenernet.org/portal/#stds_pargrp_collapse7 (accessed on 25 February 2021).

**Conflicts of Interest:** The authors declare no conflict of interest.

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
