# Peer review of "Development of the Mesoscale Model GRAMM-SCI: Evaluation of Simulated Highly-Resolved Flow Fields in an Alpine and Pre-Alpine Region"

_atmosphere, doi:10.3390/atmos12030298_

Round 1
Reviewer 1 Report
This paper presents new developments of mesoscale model GRAMM-SCI and its evaluation using WegenerNet data. One key point is the introduction of the nudging technique for temperature and humidity as well as for horizontal flow-field components, which improve substantially the performance of GRAMM-SCI published in Oettl and Verratti (2021).
The paper is well-structured and the results are discussed properly.
I have a couple of comments which would improve the quality of the paper.
1) In the conclusions the author mentions that the nesting and the novel downscaling methodologies provide robust results within reasonable computational times. I would like to hear a bit more about that. I recommend to include a comparison of the computational times before and after the modifications.
2) I believe that an useful exercise (but not essential for publication) would be to add some scatter (or qq) plots of the paired observations and predictions which may be helpful for a quick overall assessment of model performances.
Author Response
Reviewer #1:
This paper presents new developments of mesoscale model GRAMM-SCI and its evaluation using WegenerNet data. One key point is the introduction of the nudging technique for temperature and humidity as well as for horizontal flow-field components, which improve substantially the performance of GRAMM-SCI published in Oettl and Verratti (2021).
The paper is well-structured and the results are discussed properly.
I have a couple of comments which would improve the quality of the paper.
- In the conclusions the author mentions that the nesting and the novel downscaling methodologies provide robust results within reasonable computational times. I would like to hear a bit more about that. I recommend to include a comparison of the computational times before and after the modifications.
Response:
The following text has been added in section 2 (lines 134 – 138):
The limited modelling time in downscaling mode naturally speeds up the simulation by a factor of 3 compared to a nested model run with the same horizontal resolution. For instance, the modelling times for the Feldbach region were 3 h, 8h, and 10 h for the modelling domains utilizing 5000 m, 1000 m, and 200 m. In nesting mode 30 h simulation time would have been necessary for the 200 m resolution run.
- I believe that an useful exercise (but not essential for publication) would be to add some scatter (or qq) plots of the paired observations and predictions which may be helpful for a quick overall assessment of model performances.
Response:
I fully agree that scatter (or quantile-quantile) plots would provide a good and quick assessment of model performance. The reasons, why no such comparison has been undertaken are twofold: (1) in this study 24 meteorological stations are used each observing at least 2 parameters giving a total of approximately 50 parameters that one could look at. Even, if a few stations could be merged into a single scatter plot, still it would not be possible to present all parameters for all stations. Picking just a few parameters and stations would have been a rather arbitrary choice. (2) Scatter plots have little explanatory power regarding the ability of a model to cover physical processes. At the current stage of development of the GRAMM-SCI model it is very important to look at the model performance with regard to this respect. Therefore, time-series analysis was the preferred method for discussing model performance.
Reviewer 2 Report
Review for atmosphere-1105122: “Development of the mesoscale model GRAMM-SCI: Evaluation of simulated highly-resolved flow fields in an Alpine and Pre-Alpine region” By Oettl
This paper introduces recent model developments for the mesoscale model GRAMM-SCI and investigates the model performance in simulating meteorological fields, radiation fluxes, and soil characteristics. The manuscript is well structured and clearly written. The experiment to investigate the vegetation and building impacts on the simulated flow field is interesting. However, there are several places where the statements are not well supported. For example, in Ln 193, the author tries to attribute the radiation bias to cloud cover uncertainties but does not provide corresponding analysis on the simulated cloud fields. My comments are below:
Major Comments:
- Physics schemes: The section of the Model description is missing some important information regarding the physics schemes adopted in the GRAMM-SCI. What are the PBL, LSM, surface layer scheme, and microphysics schemes used in the model?
- Could fields in Section 4.1: Analysis of cloud fields are needed to investigate the radiation biases. In Ln 192, the author states that there is an overestimated cloud cover by ERA5 in the Feldbach domain. What ‘truth’ does the author use to validate the ERA5 data?
- Variations among different sites: In Figure 5 and several other figures, results are only shown for selected sites, rather than the overall mean. Can authors add justifications for this? I am also wondering if the model performance varies among the sites.
Minor Comments:
- Ln 8-9: ‘Pollutant dispersion over complex terrain remains challenging primarily because of the difficulties in correctly modelling flow and turbulence. ‘----While I agree that the accurate simulation of flow and turbulence is challenging over complex terrain, I cannot agree it is the ‘primary’ factor for air quality models to capture pollutant dispersion. Emissions resources and chemistry are equally important and challenging. Modelling ---> Modeling
- Ln 14: ‘Three one-way nested model domains’----Can the author explain why not choose two-way nested domains?
- Ln 15: ‘5.000 m’----Do you mean 5000 m? Same as in Ln 16 and Ln 143
- Ln 36 and Ln 39: Please write out ‘GRAMM-SCI’ and ‘ERA5’ where they first appear in the main text.
- Ln 44: ‘In a previous publication Oettl and Veratti (2021)’----In a previous publication by Oettl and Veratti
- Ln 67: ‘only those changes in the model will be outlined that have been implemented since then’----Please reorganize the sentence.
- Ln 123: ‘what would justify’---‘that would justify’
- Ln 137: What is GRAL? I think it is reasonable to add some information regarding GRAL since the results regarding the buildings and vegetation's impacts on microscale flows are of interest to the research community.
- Figure 1: Do the two domains cover the two regions in WegenerNet, respectively? Please denote the two regions, Feldbatch and Johnbach in Figure 1.
- Figure 2: What do H, T and HTK mean on the weather map?
- Figure 3 and Figure 4: I suggest putting the two figures side by side in one figure
- Table 2: I don’t think it is necessary to keep so many digits for the lat and lon; I suggest changing the abbreviation for air temperature from at to Tair, or some other terms the author prefers.
- Equations (3)-(5): The meanings of n and arccos are missing.
- Figure 6: What is the depth of the first soil layer? Is it consistent with the observational data (0.2 m)?
- Figure 8: The warm bias between 15th and 17th is really interesting. Are there more mixing in the model?
- Figure 10: It is not easy to tell the four lines with similar patterns and colors. Please consider changing the line colors.
Author Response
This paper introduces recent model developments for the mesoscale model GRAMM-SCI and investigates the model performance in simulating meteorological fields, radiation fluxes, and soil characteristics. The manuscript is well structured and clearly written. The experiment to investigate the vegetation and building impacts on the simulated flow field is interesting. However, there are several places where the statements are not well supported. For example, in Ln 193, the author tries to attribute the radiation bias to cloud cover uncertainties but does not provide corresponding analysis on the simulated cloud fields. My comments are below:
Major Comments:
Physics schemes: The section of the Model description is missing some important information regarding the physics schemes adopted in the GRAMM-SCI. What are the PBL, LSM, surface layer scheme, and microphysics schemes used in the model?
Response:
Unlike other mesoscale models (e.g. WRF), GRAMM-SCI has no option for choosing different physical parameterizations, with the exception of turbulence. A detailed description of all the schemes would go far beyond the scope of this work. The reader is therefore referred to the model documentation, which can be downloaded here (the corresponding quotation is already provided in the manuscript): https://www.researchgate.net/profile/Dietmar_Oettl/publications
As mentioned in the manuscript, a more detailed description of the model physics is given in a recent publication (Oettl and Verratti, 2021), which is not repeated in the manuscript.
As said this, a short paragraph has been added in the manuscript (lines 108 – 115) in order to give a very brief overview about the physics schemes:
For the turbulence closure a zero-order mixing length model is utilized (e.g. Pielke, 1984), where the turbulent viscosity depends on the local Gradient-Richardson number. The radiation model used in GRAMM-SCI is based on the work of Somieski (1988). Surface fluxes are calculated depending on stability and roughness-length according to relations proposed by Businger (1971). Microphysical schemes for computing clouds and precipitation are not yet available in GRAMM-SCI. Cloud and snow cover are retrieved from ERA5 reanalysis data and used diagnostically for adjusting the soil heat flux, albedo, and radiation Oettl and Veratti (2021).
Could fields in Section 4.1: Analysis of cloud fields are needed to investigate the radiation biases. In Ln 192, the author states that there is an overestimated cloud cover by ERA5 in the Feldbach domain. What ‘truth’ does the author use to validate the ERA5 data?
Response:
I agree that the argumentation was obscure in the manuscript, because it was not clear for the reader that GRAMM-SCI does not compute any cloud formation but rather takes the cloud cover as provided by ERA5. Therefore, I’ve rewritten the paragraph in the following way (lines 211 – 214):
On the 13th Oct. GRAMM-SCI clearly underestimates the global radiation in the Feldbach domain. As mentioned in section 2, cloud formation is not calculated in GRAMM-SCI yet. Therefore, the underestimation of global radiation can only be due to an overestimated cloud cover by ERA5.
Variations among different sites: In Figure 5 and several other figures, results are only shown for selected sites, rather than the overall mean. Can authors add justifications for this? I am also wondering if the model performance varies among the sites.
Response:
As to the overall mean comparison, the following paragraph has been added before section 4.1 (lines 203 – 207):
In some studies the overall mean of observed quantities is used for demonstrating mesoscale model performance. This seems to be particularly useful as long as the horizontal resolution in the modelling is still rather coarse (≥ 1 km). However, the fine horizontal resolution used in this study allows for direct comparison with observations at specific locations, which was therefore the preferred method.
As to the performance variation among sites: Regarding wind speed, Figure 8 illustrates the variation in performance for all stations using NMSE and BIAS as quality indicators. In the revised manuscript a similar evaluation has been undertaken for wind direction (included in Figure 8). Regarding soil temperature it has already been stated in section 4.1 that modelled values vary hardly among the stations, while observations reveal higher spatial variability. In contrary, soil moisture content was found to be quite homogenous for both modelling and observations. Global and net radiation were only observed at 5 stations in the Johnsbach region. In the Feldbach domain only one station was available for comparing global radiation (which is shown in Figure 4). Little variation was found regarding net radiation, i.e. the comparisons shown for stations WN501 and WN505 are well representing the other three stations (WN502, WN503 and WN504). As already pointed out in the manuscript, the mean bias over all 6 stations for the global radiation was higher than for the net radiation. A large bias is found for the station WN505 (-71 W/m²). The station is located on a mountain peak at 2191 m making the simulated global radiation very sensitive to the exact position of the station within the model’s topography. That means that even a slight dislocation between the station and the mountain peak as mapped by the model immediately leads to an erroneous orientation of the model’s surface. The text written in Italic style has been added in section 4.1 (lines 217 – 222). In addition, results for the station WN505 are included in Figure 4.
Minor Comments:
Ln 8-9: ‘Pollutant dispersion over complex terrain remains challenging primarily because of the difficulties in correctly modelling flow and turbulence. ‘----While I agree that the accurate simulation of flow and turbulence is challenging over complex terrain, I cannot agree it is the ‘primary’ factor for air quality models to capture pollutant dispersion. Emissions resources and chemistry are equally important and challenging. Modelling ---> Modeling
Response:
I agree. The sentence has been deleted.
Ln 14: ‘Three one-way nested model domains’----Can the author explain why not choose two-way nested domains?
Response:
Currently, GRAMM-SCI has no option for two-way nesting.
Ln 15: ‘5.000 m’----Do you mean 5000 m? Same as in Ln 16 and Ln 143
Response:
Yes. The numbers have been changed.
Ln 36 and Ln 39: Please write out ‘GRAMM-SCI’ and ‘ERA5’ where they first appear in the main text.
Response:
The full names are provided in the manuscript (line 34 and 37), where they appear first.
Ln 44: ‘In a previous publication Oettl and Veratti (2021)’----In a previous publication by Oettl and Veratti
Response:
Changed.
Ln 67: ‘only those changes in the model will be outlined that have been implemented since then’----Please reorganize the sentence.
Response:
The sentence has been rephrased (lines 66 – 67).
Ln 123: ‘what would justify’---‘that would justify’
Response:
Changed.
Ln 137: What is GRAL? I think it is reasonable to add some information regarding GRAL since the results regarding the buildings and vegetation's impacts on microscale flows are of interest to the research community.
Response:
I’ve added more information about GRAL-SCI (lines 151 – 154). Corresponding scientific publications as well as the GRAL-SCI documentation have already been quoted in the manuscript.
Figure 1: Do the two domains cover the two regions in WegenerNet, respectively? Please denote the two regions, Feldbach and Johnsbach in Figure 1.
Response:
The domains have been denoted in Figure 1 in the revised manuscript.
Figure 2: What do H, T and HTK mean on the weather map?
Response:
The corresponding English translations have been added in the figure caption (H = high pressure; T = low pressure; HTK = upper-level low)
Figure 3 and Figure 4: I suggest putting the two figures side by side in one figure
Response:
Both figures have been put side by side.
Table 2: I don’t think it is necessary to keep so many digits for the lat and lon; I suggest changing the abbreviation for air temperature from at to Tair, or some other terms the author prefers.
Response:
The digits have been set to 5, which corresponds to about 1m accuracy for the given latitude of 47 deg. The abbreviation for air temperature, soil temperature, and soil moisture have been changed, too.
Equations (3)-(5): The meanings of n and arccos are missing.
Response:
n is the number of data points (explanation has been added), and arccos is the inverse of the cosine function. As arccos is a widespread notation used in English, no explanation is needed in my opinion.
Figure 6: What is the depth of the first soil layer? Is it consistent with the observational data (0.2 m)?
Response:
The modelled soil temperature is calculated by a linear interpolation between the 5th and 6th soil layer, which are located 18 cm and 42 cm below the surface. It has been clarified in the text (lines 236 - 237).
Figure 8: The warm bias between 15th and 17th is really interesting. Are there more mixing in the model?
Response:
Frankly spoken, I don’t know. Wrong mixing could be one issue, but there are other candidates as well, such as an underestimation of the latent heat flux for instance.
Figure 10: It is not easy to tell the four lines with similar patterns and colors. Please consider changing the line colors.
Response:
I’m bit old-fashioned regarding the usage of colors. I’d like to leave this up to the editor. Note, that Figure 10 (Figure 9 in the revised manuscript) would be the only colored figure in the paper.
Reviewer 3 Report
the paper concerns basically the analysis of the performances of this model in the simulation of mesoscale and submeso features of the boundary layer in complex terrain.
My overall opinion is that the argument could be more proper for a technical report, but the editor shall decide about this point.
I do not question about the discussion on the statistical indexes, that turns out to be exhaustive although boring (for a scientific paper) from my point of view.
A major point concerns the submeso motion and the related variability of the wind field (line 385 and following lines). This aspect is of overwhelming importance especially in the context of pollutant dispersion (quoted in the first line of the summary), and is explicitly stated that the present model cannot deal with that phenomenon. Moreover, no words are deserved to the estimation of the velocity variances (both in the case of absence or presence of small frequency fluctuations), which are crucial in dispersion evaluations.
Thus, a mandatory indication for publication is to drop any direct reference to the dispersion, and possibly to state that the model does only evaluate first order moments of the meteorological variables and thus is not directly applicable to dispersion studies.
Minor comments:
figures and tables are badly formatted (see for instance eq 7 table 3 fig 10)
Author Response
the paper concerns basically the analysis of the performances of this model in the simulation of mesoscale and submeso features of the boundary layer in complex terrain. My overall opinion is that the argument could be more proper for a technical report, but the editor shall decide about this point. I do not question about the discussion on the statistical indexes, that turns out to be exhaustive although boring (for a scientific paper) from my point of view.
A major point concerns the submeso motion and the related variability of the wind field (line 385 and following lines). This aspect is of overwhelming importance especially in the context of pollutant dispersion (quoted in the first line of the summary), and is explicitly stated that the present model cannot deal with that phenomenon. Moreover, no words are deserved to the estimation of the velocity variances (both in the case of absence or presence of small frequency fluctuations), which are crucial in dispersion evaluations.
Thus, a mandatory indication for publication is to drop any direct reference to the dispersion, and possibly to state that the model does only evaluate first order moments of the meteorological variables and thus is not directly applicable to dispersion studies.
Response:
I do fully agree with that statement. The very first sentence in the summary, which refers to pollution dispersion, has been deleted. Apart from this sentence, air-pollution modelling has mentioned three times in the text. All these statements have been deleted.
Minor comments:
figures and tables are badly formatted (see for instance eq 7 table 3 fig 10)
Response:
I’m not quite sure whether I understood the point of revierwer #3. It seems as eq. 7, table 3, and fig. 10 are positioned at the left hand side, while the corresponding text is orientated to the right. If this is the issue, then it has been caused by the pre-editing process of the journal.
Round 2
Reviewer 2 Report
Thank the author for addressing my comments. I recommend publication.